# Evolution of basic human values orientations: An application of monitoring changes in cluster solutions

**Muhammad Atif**[1,2]*, **Muhammad Shafiq**[3], **Muhammad Farooq**[1], **Gohar Ayub**[4], **Mujeeb Hussain**[5], **Muhammad Waqas**[6]

**1** Department of Statistics, University of Peshawar, Peshawar, Pakistan, **2** Institute of Statistics, University of Natural Resources and Life Sciences, Vienna, Austria, **3** Institute of Numerical Sciences, Kohat University of Science and Technology, Kohat, Pakistan, **4** Department of Mathematics and Statistics, University of Swat, Swat, Pakistan, **5** Government College Peshawar, Peshawar, Pakistan, **6** University of Swabi, Swabi, Pakistan

* dratif96@gmail.com.pk

**Data Availability Statement:** The data that support the findings of this study are openly available in "European Social Survey" at http://doi.org/10.

## Abstract

This study enumerates the evolution of basic human values orientations and the dynamic relationship between them, computed from Schwartz's value survey conducted in European nations. For this purpose, eight datasets related to the human value scale were extracted from the European Social Survey; each corresponds to a single round conducted cross-sectionally every two years since 2001. Change detection algorithm was implemented to the cluster solutions of temporal datasets, and the evolution of important clusters was traced. Finding of the study reveals that Universalism and Benevolence values are on the rise in European societies in the last couple of decades. Most of the European inhabitants believe in the smooth group functioning and form the organismic needs of cooperation. The people prefer anxiety-free life, and love for nature, environment, humanity, and kindness to other beings in society are essential constructs for them. They avoid self-centred behaviour and prefer social physiognomies.

## Introduction

Over the course of last decade, an abrupt development in the field of science and technology have extreme affects on human life. In the past days, a nation's traditions, norms, values, and social interactions were considered to be conservative and resistant to the influences of other cultures. However, an abrupt development in the field of science greatly affects the human values and norms. Value is a conception explicit or implicit, distinctive of an individual or characteristics of a group of those desirable traits which influence the selection from available modes and ends of action. People in the individual mode or collectively in the society evaluates other peoples and events in the light of values [1, 2]. In simple words the notion of basic human values guides us to perform certain actions while considering the interactions with other human beings in a society. They are the guiding principles essential for positive behavior

21338/NSD-ESS-CUMULATIVE. [doi:10.21338/
NSD-ESS-CUMULATIVE.]

**Funding:** The author(s) received no specific
funding for this work.

**Competing interests:** The authors have declared
that no competing interests exist.

in our daily life activities. The social scientists acknowledge the values as beliefs that explain and justify the attitudes, actions, and opinions of individuals [3]. Though in past centuries the paradigm of human values much suffered from the absence of empirical models, agreed-upon theories, and philosophical frameworks to measure them [4, 5]. Yet recent development in building theoretical backgrounds and methodological models for quantifying the human values recuperate the modern-day research in this area [6, 7].

Several models such as [8–10] have been developed for enumerating the relationship between human values and its impacts on social and personal organization in a community. However, the most popular and widely used method in recent years is Schwartz value survey (SVS) scale [6]. Because the Schwartz value theory not only quantifies the basic human values but also explains the relationship between these values i.e. it demonstrates how these values are interconnected and influence one another. According to Schwartz, there are ten basic human values, each described in terms of their respective goals. These values are designed in such a manner that they encompass all the values that were practised around the globe. That is it covers the contents found in earlier values from different cultures, religions, and philosophical models. The ten motivationally distinct values are Benevolence (BE), Universalism (UN), Self-direction (SD), security (SE), Confirmatory (CO), Hedonism (HE), Achievements (AC), Traditions (TR), Stimulation (ST), and Power (PO). Respondents of the survey are asked to rate the importance of each item as a guiding principle in their life on a 9-point scale labeled as 7 (of supreme importance), 6 (very important), 5, 4 (unlabeled), 3 (important), 2, 1 (unlabeled), 0 (not important), and -1 (opposed to my values). The raw ratings of each value item are calculated on a non-symmetrical scale to motivate the respondents to think about each of the items. The average of the raw ratings assigned to the items associated with each value is designated as a priori indicator of that value.

In addition to quantifying the raw ratings of each human value, the Schwartz theory also describes the dynamic relationship amongst these values. That is it explains how the actions for pursuing one particular value affect the pursuit of other values. For example, desiring success for oneself can leads to obstructing the actions required for enhancing the welfare of others who are in need of one's help in the society. Similarly, seeking the adaptation of novelty and change potentially damages the actions required for preserving time-honoured customs and traditions. The individual picking amongst these alternate actions leads toward practical, social, and psychological constructs in the society and decides their integrated actions. Undoubtedly individuals can pursue contending values, but not in a single act, they do so at different times in different backgrounds [11].

The Communal values play an important and critical role in the development and building of a future that everybody wants to experience. Because the values have a significant impact on our decisions while taking into consideration the interactions with others members and building an internal cohesion in a society. Any changes in individual human values bring a drastic impact on societal ethical judgment and actions. Although according to some researchers values are consistent over time [12, 13], yet it changes gradually. Societal acclimatization to exogenous factors such as interactions with other societies, wealth, advancement in science and technology, demographics, and epidemic factors leads to deviations in cultural values orientations [14, 15]. Obviously dominant cultures are not completely coherent, and sub-groups within society are prone to adopt conflicting values by interacting with other cultures.

Dewangan [16] debated the impact of basic human values on the different aspects of society. He claims that it touches upon every aspect of our life ranging from political, economic, business, education, humanity, technology, ethical values, to cultural values. Wolf et al. examined the significance of human values in venturing the COVID-19 pandemic. The study's results reveal that human values and their importance amongst fellow citizens are potential factors for

tackling the COVID-19 crisis. People who focus on self-centered values such as self-transcendence and security are probably more compliant with COVID-19 behavioral guidelines. Furthermore, perceiving that others share one's values is likely to evoke a feeling of connectedness that may be essential in encouraging collaborative measures to control the pandemic [17].

In recent years, one of the most exciting topics is the study of diversity and cultural changes over time. Certain computer-based data-driven approaches detect and demonstrate a shift in people behavior, with clear implications for tracking culture changes [18–20]. To our knowledge, no study was conducted so far that quantifies the basic human values and investigates the effect of alien cultures on these values over time. This study was conducted with the aim to quantify the basic human values of European nations at discrete points and monitor the changes in these values. Also, this study aims to investigate the dynamic relationship between human values and identify the emerging traits in the European societies. We extract multiple datasets on the SVS instrument to achieve the aforementioned goals.

## Objectives

In the light of these goals, this study was conducted to achieve the following objectives:

- To evaluate the scores for the ten basic human values from temporal datasets

- To monitor the evolution of human values by tracing changes in cluster solutions of temporal datasets

- To examine the dynamic relation between the emerging values in the society

## Materials and methods

The datasets used in this paper were extracted from European Social Survey (ESS), and clusters' evolution was traced to identify transition in values orientations. Most of the European countries host a large community of immigrants from across the globe, with 5.3% of its population [21]. Exposure to such a large number of societies with diverse cultures across the globe will have an impact on societal ethics, and a segment of the community will accustom values from these migrants.

### Datasets

In this paper, we extract eight datasets on SVS scale from ESS cumulative data wizard, each corresponds to a single round carried out cross-sectionally in years 2002, 2004, 2006, 2008, 2010, 2012, 2014, and 2016 respectively [22] and can be downloaded from the URL https://ess-search.nsd.no/CDW/ConceptVariables. The ESS conduct academically driven surveys every two years since its foundation in 2001 to evaluate the attitudes, beliefs, and behavior pattern of European nations. The human value scale has been included in every ESS round since 2002 to date. The ESS "Basic Human Values" questionnaire comprises of a well-established 21 items scale, designed by Schwartz (1992) (see Table 1 **Items in SVS questionnaire and their associated human values**). These items organize ten motivationally distinct human values and specify the dynamics of respondents according to their value score orientations [23]. Table 1 given in S1 Appendix includes these 21 items and their associated human values.

### Methods

One of the most commonly used data mining tasks is clustering, which widely studies the similarity between features amongst data records [24]. In simple words, the aim is to segregate

**Table 1. Items in SVS questionnaire and their associated human values.**

| Human Value | Number | Item |
|---|---|---|
| BENEVOLENCE | 12 | It's very important to him to help the people around him. He wants to care for other people. |
| | 18 | It is important to him to be loyal to his friends. He wants to devote himself to people close to him. |
| UNIVERSALISM | 03 | He thinks it is important that every person in the world be treated equally. He wants justice for everybody, even for people he doesn't know. |
| | 08 | It is important to him to listen to people who are different from him. Even when he disagrees with them, he still wants to understand them. |
| | 19 | He strongly believes that people should care for nature. Looking after the environment is important to him. |
| SELF-DIRECTION | 1 | Thinking up new ideas and being creative is important to him. He likes to do things in his own original way. |
| | 11 | It is important to him to make his own decisions about what he does. He likes to be free to plan and to choose his activities for himself. |
| STIMULATION | 06 | He likes surprises and is always looking for new things to do. He thinks it is important to do lots of different things in life. |
| | 15 | He looks for adventures and likes to take risks. He wants to have an exciting life. |
| HEDONISM | 10 | Having a good time is important to him. He likes to"spoil" himself. |
| | 21 | He seeks every chance he can to have fun. It is important to him to do things that give him pleasure. |
| ACHIEVEMENT | 04 | It is very important to him to show his abilities. He wants people to admire what he does. |
| | 13 | Being very successful is important to him. He likes to impress other people. |
| POWER | 02 | It is important to him to be rich. He wants to have a lot of money and expensive things. |
| | 17 | It is important to him to be in charge and tell others what to do. He wants people to do what he says. |
| SECURITY | 05 | It is important to him to live in secure surroundings. He avoids anything that might endanger his safety. |
| | 14 | It is very important to him that his country be safe from threats from within and without. He is concerned that social order be protected. |
| CONFORMITY | 07 | He believes that people should do what they're told. He thinks people should follow rules at all times, even when no-one is watching. |
| | 16 | It is important to him always to behave properly. He wants to avoid doing anything people would say is wrong. |
| TRADITION | 09 | He thinks it's important not to ask for more than what you have. He believes that people should be satisfied with what they have. |
| | 20 | Religious belief is important to him. He tries hard to do what his religion requires. |

groups with similar traits and assign them into clusters. In cumulative datasets and data streams, clustering is mainly dominated by out-of-date historical information. However, in most real-life applications, the recent data records are considered more preferable and substantial. Hence clustering the streaming datasets over a sliding window becomes a natural and popular choice of modelling [25]. In this paper, we implement a sliding window model of size $w = 3$, and data records that fall in the interval $[t_i − w + 1]$ were maintained in each window pane. The data records older than $t_i − w + 1$ were discarded at each iteration of the sliding window model. As a result of this implementation, the sliding window model generates $n-w+2$ window panes of cumulative datasets, where $n$ represents total number of time points in the temporal dataset. These window panes of cumulative datasets were clustered separately which generate a sequence of clustering solutions, i.e. one cluster solution from each corresponding window pane.

- **Clustering Algorithm**: In order to extract clusters from the window panes, the $k$-means clustering algorithm was implemented separately to each pane. The $k$-means was decided due to its speed and capacity to operate with large datasets. The Euclidean distance between values scores of participants was used as a dis-similarity metric [26].

- **Number of clusters $k$**: The choice of an optimal number of clusters was decided by silhouette width, a measure of internal validity. The silhouette statistic measure the extent of similarity between the objects belonging to the same cluster compared to other clusters. Its value ranges from −1 to +1, where a value close to +1 indicates higher similarity between the objects of its own cluster and inadequately matched to adjacent clusters. If most objects have

a high value, then the clustering configuration is appropriate [27]. We use the *factoextra()* and *Nbclust()* packages from R-software to estimate the optimal number of clusters. (for more details see S1 Fig and Fig 1).

Multidimensional scaling (MDS) was used for the graphical representation of relation amongst human values in important clusters. MDS refers to a family of models that analyzes the proximity between objects in a dataset. Proximity refers to the similarity or dissimilarity between the objects. Information possessed by a set of data is expressed by a set of points in space. These points are organized in such a way that their geometrical distance reflects the empirical relationships in the data. So, it is a visual presentation of (dis)similarities between sets of objects in a data. Scores on the objects that are more similar (having shorter distances) are closer together on the chart than objects that are dissimilar (having longer distances). Furthermore, MDS also suits as a dimension reduction technique for high-dimensional data. The similarity between values scores in plotted in a two dimensional Cartesian plan.

To trace the evolution of clusters in cumulative datasets [28], developed a state-of-the-art algorithm known as MONIC framework. This framework receives the temporal datasets as input and cluster them at successive discrete time points. The clustering solution extracted at

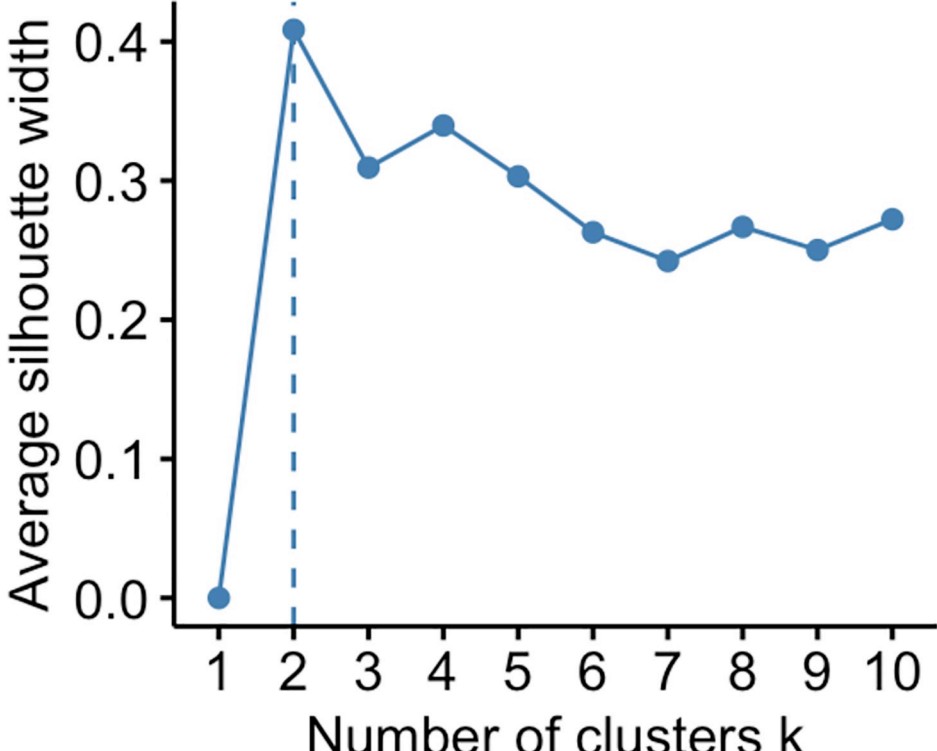

**Fig 1. Optimal number of clusters from silhouette statistics.**

time-point $t_i$ is compared with the clustering solution at $t_{i-1}$, and the transition of clusters are monitored over time. In the context of this framework, the transition of clusters is the change experienced by a cluster that emerged at later time point $t_i$ in reference to the cluster obtained at an earlier time-point $t_{i-1}$. Such changes include the external and internal transition of clusters.

External transition includes migration of some elements from one cluster to another, still a part of the same cluster, disappearing of old clusters, and emerging of new ones. These transitions are classified into five categories i.e. survived, split, merge, emerged, and died candidates. While on the other hand, internal transition refers to changes in the form of survived clusters i.e. size, density, and location. We developed an R package for this algorithm with the name **clusTransition**, which can be downloaded from the url https://CRAN.R-project.org/package= clusTransition [29].

[30] presents a detailed analysis of the algorithms used for monitoring and tracing the evolution of cluster solutions in temporal datasets. To illustrate the applications and importance of monitoring the cluster's development over time, they used several published datasets. We used the values scores computed from SVS to monitor the changes in human values. But before any statistical analysis of the values, it is important to correct them for scale. Below is the process of scale correction [31]:

- First of all, compute a score for each of ten values by averaging the items that index it.

- Compute overall average for all 21 items, called MRAT.

- The centered score is the difference between individual value score and MRAT.

## Results and discussion

A sliding window model of size *w = 3* was implemented, which generate 7 window panes of cumulative datasets. Table 2 below demonstrates the cumulative number of participants, after removing cases with missing observations, and the optimal number of clusters in each window pane.

Fig 2 below demonstrates the consistency of the cluster solutions at subsequent time points in data streams. The cluster's consistency is investigated using the survival and pass-forward ratios of cluster solutions. A low survival and pass-forward ratios indicate a drastic change in the clustering solutions of the underlying data stream. In such cases, the old clusters are famished resulting in newly emerged clusters. The sub-plot B and C represents the number of newly emerged and disappeared clusters at corresponding time points. The sub-plots A and C represents survival and pass-forward ratios. It is evident from Fig 2 that the survival ratio and pass-forward ratio are reasonably low, and mostly the clusters disappear at successive time points. As a result, newly born clusters were detected by the algorithm at each time point. The

**Table 2. Number of participants and optimal number of clusters.**

| Window panes | $D_1$ | $D_2$ | $D_3$ | $D_4$ | $D_5$ | $D_6$ | $D_7$ |
|---|---|---|---|---|---|---|---|
| | (2002) | (2002–2006) | (2004–2008) | (2006–2010) | (2008–2012) | (2010–2014) | (2012–2016) |
| n | 2833 | 8619 | 8537 | 8698 | 8740 | 9014 | 8835 |
| k | 4 | 4 | 3 | 3 | 3 | 3 | 3 |

The implementation of the sliding window model generates 7 panes of cumulative datasets. The second row shows the cumulative number of participants in respective surveys. Whereas the third row represents the optimal number of clusters in each window pane. The optimal number of clusters is calculated using the silhouette statistics.

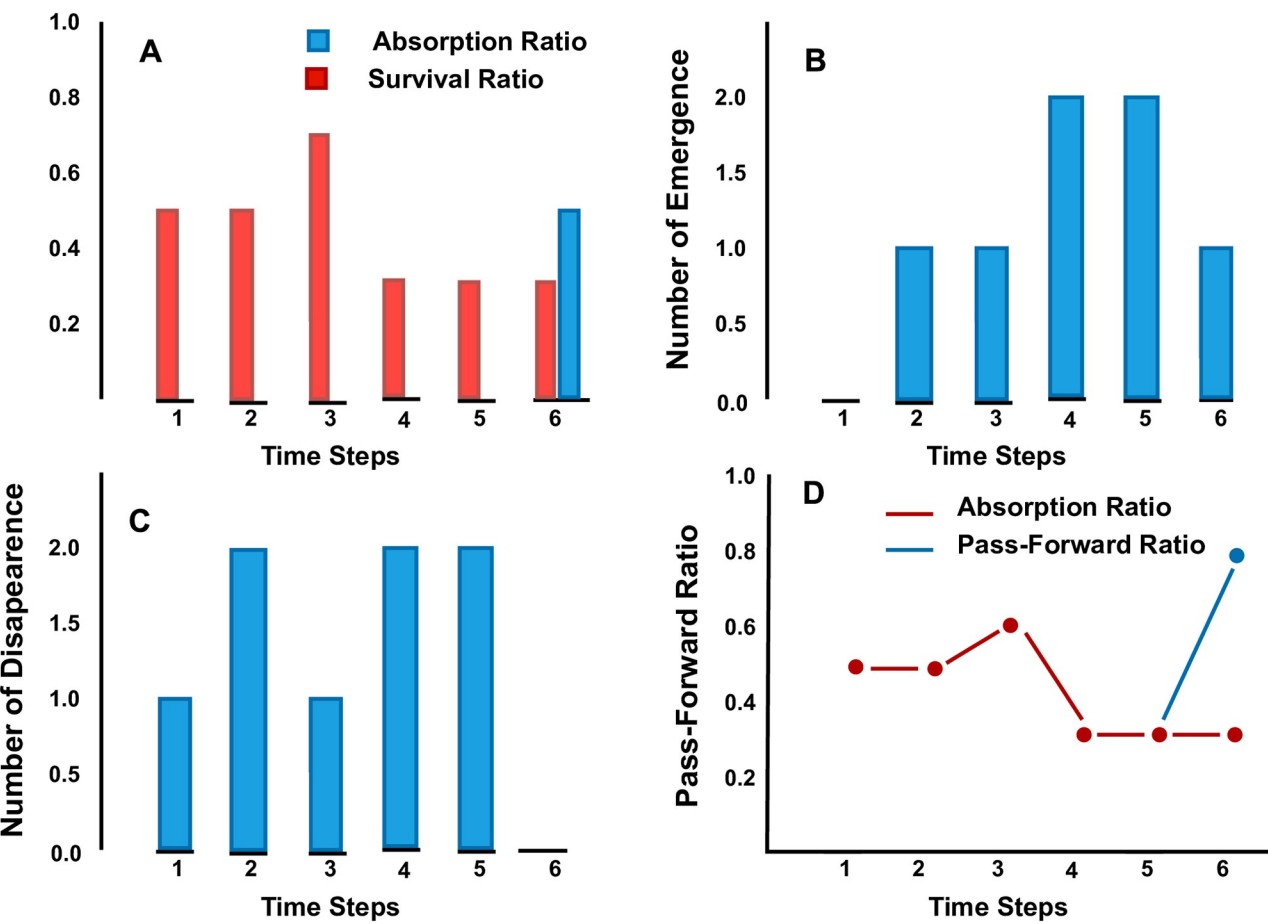

**Fig 2. Changes adopted by clusters at respective time points.**

deaths of so many clusters is an indication that the human values are changing drastically and the people are adopting new values.

Results of cluster evolution's are summarized in Fig 3 below, which demonstrates that two clusters $C_{11}$ and $C_{12}$ survived over time. The first imperative cluster was $C_{11}(C_{11} \rightarrow C_{22} \rightarrow C_{32} \rightarrow C_{42})$ that emerged at $t_1(2002)$ and survived until $t_4(2006, 2010)$. Though, the cluster survived till 2010, but experienced internal transition and became more diffused eventually disappeared at time-point $t_5$. The second vibrant cluster was $C_{12}(C_{12} \rightarrow C_{24} \rightarrow C_{33} \rightarrow C_{41} \rightarrow C_{52} \rightarrow C_{63} \rightarrow C_{71})$ which survive through the entire time span. This was the most important cluster because not only it survives over time but also turns out to be denser. Mostly the new respondents of SVS surveys over the years joins this cluster. The shift in location was observed for this cluster at time-point $t_2$ and $t_3$, and afterward, remain stable. The first external transition was experienced in the cluster $C_{14}(C_{14} \rightarrow \{C_{21}, C_{23}\})$ which split into two clusters and ultimately disappeared. The algorithm also detects a cluster $C_{61}$ that emerged at $t_6(2010, 2014)$ and pass-forward while absorbing elements of the cluster $C_{62}$. Apart from these external and internal transitions, the algorithm also detects several clusters that emerged but soon disappeared. The algorithm detected shift in location at time point $t_2$ and $t_3$ only, afterwards the cluster remain stable.

Table 3 given below presents the scores of ten basic values and their MART for the first imperative cluster $C_{12}$ that survived through time. The centred scores for Power and Security

**Time Points**

**Fig 3. Cluster's evolution over time.** The nodes represents the clusters whereas the edges represents the transition experienced by respective clusters. The first subscript in $C_{ij}$ represents the time points, whereas the second subscript represents the cluster.

decreased gradually, whereas the centred score for Benevolence and Universalism gradually increased through time. The power value is self-centred that focuses on social status, prestige, dominance over people and resources. Similarly, the security value also primarily serves as individual interests such as self-security and self-stability in society. On the other hand, Benevolence and universalism focus on social goals such as preserving and enhancing the welfare of others, honesty, tolerance, and welfare of other people and love for nature.

In order to investigate the dynamic relationship amongst the basic values, their scores are visualized in a Cartesian space using multidimensional scaling. Fig 4 below demonstrates the multidimensional scaling of clusters ($C_{12} \rightarrow C_{24} \rightarrow C_{33} \rightarrow C_{41} \rightarrow C_{52} \rightarrow C_{63} \rightarrow C_{71}$). This cluster survive from time point $t_1$ to time point $t_7$ over time. On first dimension the Universalism and Benevolence have high similarity and emerged together. These values focuses on the smooth group functioning and from the organismic need for affiliation. The people realizes that failure to protect the natural environment will lead to the destruction of the resources on which life depends. The people are broadminded and believe in social justice, equality, and protecting the environment. Similarly, Self-Direction values based on independent thoughts and freedom in choosing own goals. These values are anxiety free, as people believe in taking care of nature, environment, humanity and kindness to other beings. Similarly, the values Hedonism, Stimulation, and Achievement contributed to the second dimension. These values primarily focus on openness to change in society, sensuous gratification for oneself, and enjoying life. Conservative values like Traditions and confirmatory, which focus on harming others, violating social norms, and acceptance of ideas based on culture receives similar ranks. The scores on these ranks are decreasing in this cluster. This cluster produce almost similar pattern to the one explained by [3]. The scores for social focus values as well as size of this cluster increase gradually as time progresses. This is a clear indication that inhabitants of European society adore living anxiety-free life favouring social traits over self-centred values.

Table 4 given below presents the values scores and their MART for the second imperative cluster ($C_{11} \rightarrow C_{22} \rightarrow C_{32} \rightarrow C_{42}$) over time. This cluster experience internal transition becoming

**Table 3. Values scores and MART for cluster $C_{12}$ at respective time points.**

| Window Panes | $D_1$ | $D_2$ | $D_3$ | $D_4$ | $D_5$ | $D_6$ | $D_7$ |
|---|---|---|---|---|---|---|---|
| | (2002) | (2002,2006) | (2004,2008) | (2006,2010) | (2008,2012) | (2010,2014) | (2012,2016) |
| Conformity | .67 | .17 | .11 | -.15 | .21 | .19 | .22 |
| | (0.75) | (0.96) | (0.99) | (0.86) | (0.57) | (0.95) | (1.01) |
| Tradition | -3.7 | -.02 | .26 | -.07 | .16 | -.22 | -.06 |
| | (0.66) | (1.08) | (0.77) | (0.80) | (0.54) | (1.10) | (0.72) |
| Benevolence | .08 | 0.09 | .13 | .14 | .25 | .52 | .71 |
| | (0.95) | (1.96) | (0.81) | (0.95) | (0.621) | (1.01) | (0.85) |
| Universalism | .65 | .73 | .85 | 1.03 | 1.11 | 1.14 | 1.19 |
| | (0.45) | (0.81) | (0.55) | (0.60) | (0.41) | (0.49) | (0.69) |
| Self-Direction | -.70 | -.52 | -.51 | -.09 | -.14 | -.12 | .05 |
| | (0.77) | (1.05) | (0.89) | (0.80) | (0.71) | (0.83) | (0.89) |
| Stimulation | 1.17 | .59 | .12 | .21 | .08 | .29 | .17 |
| | (0.86) | (0.96) | (0.82) | (0.69) | (0.42) | (0.79) | (0.61) |
| Hedonism | .26 | .10 | .10 | .20 | .11 | .01 | -.08 |
| | (1.08) | (1.07) | (0.63) | (0.79) | (0.37) | (0.71) | (0.49) |
| Achievement | -.08 | .10 | -.02 | .13 | .12 | .23 | -.06 |
| | (0.74) | (0.76) | (0.54) | (0.55) | (0.38) | (0.83) | (0.79) |
| Power | .84 | .17 | -.15 | -.14 | -.07 | .32 | .13 |
| | (1.22) | (1.16) | (0.90) | (0.81) | (0.55) | (0.83) | (0.75) |
| Security | .44 | .37 | .36 | .34 | .29 | -.13 | -.03 |
| | (.681) | (.712) | (.695) | (.693) | (.682) | (.673) | (.689) |
| Overall | 2.74 | 2.78 | 2.77 | 2.74 | 2.67 | 2.65 | 2.68 |
| | (.654) | (.734) | (.734) | (.753) | (.708) | (.692) | (.730) |
| $n_i$ | 947 | 2936 | 3191 | 3290 | 3277 | 3527 | 3337 |

$(C_{12} \rightarrow C_{24} \rightarrow C_{33} \rightarrow C_{41} \rightarrow C_{52} \rightarrow C_{63} \rightarrow C_{71})$. This cluster survive from time point $t_1$ to time point $t_7$

more diffused than its ancestor clusters at each successive time point. Eventually the cluster disappear at time point $t_4$(2006, 2010). The values scores on power and achievement increases gradually, whereas scores on Universalism, Hedonism, Stimulation and Benevolence decrease over time. This suggests that this particular segment of people focuses on social status and prestige, control over people and resources, and personal success. While they ignore traits like novelty, and challenge in life, preserving and enhancing the welfare of others, and protection for the welfare of all people and for nature.

Fig 5 below demonstrates the multidimensional scaling of $(C_{11} \rightarrow C_{22} \rightarrow C_{32} \rightarrow C_{42})$ clusters. On the first dimension, Hedonism and Stimulation values have higher similarities and emerged together. Hedonism values derive from organismic needs and the pleasure associated with satisfying them such as enjoying life, self-indulgence, and novelty in life. Clearly, these values primarily focus on pleasure or sensuous gratification for oneself which is an obvious symptom of anxiety-based constructs. Because if the person in individual or collectively in the society fails to achieve these gratifications or excitement, they start suffering depression and anxiety. In the second dimension the Power and Conformity values emerged together receiving same scores from the participants. The Power value along with Conformity may be a transformations of individual needs for dominance and control. These values derive the individual, s inclinations towards disrupting smooth interaction and group functioning, valuing authority, wealth, and social power. Similarly Achievements, Self-Direction, and Security move together

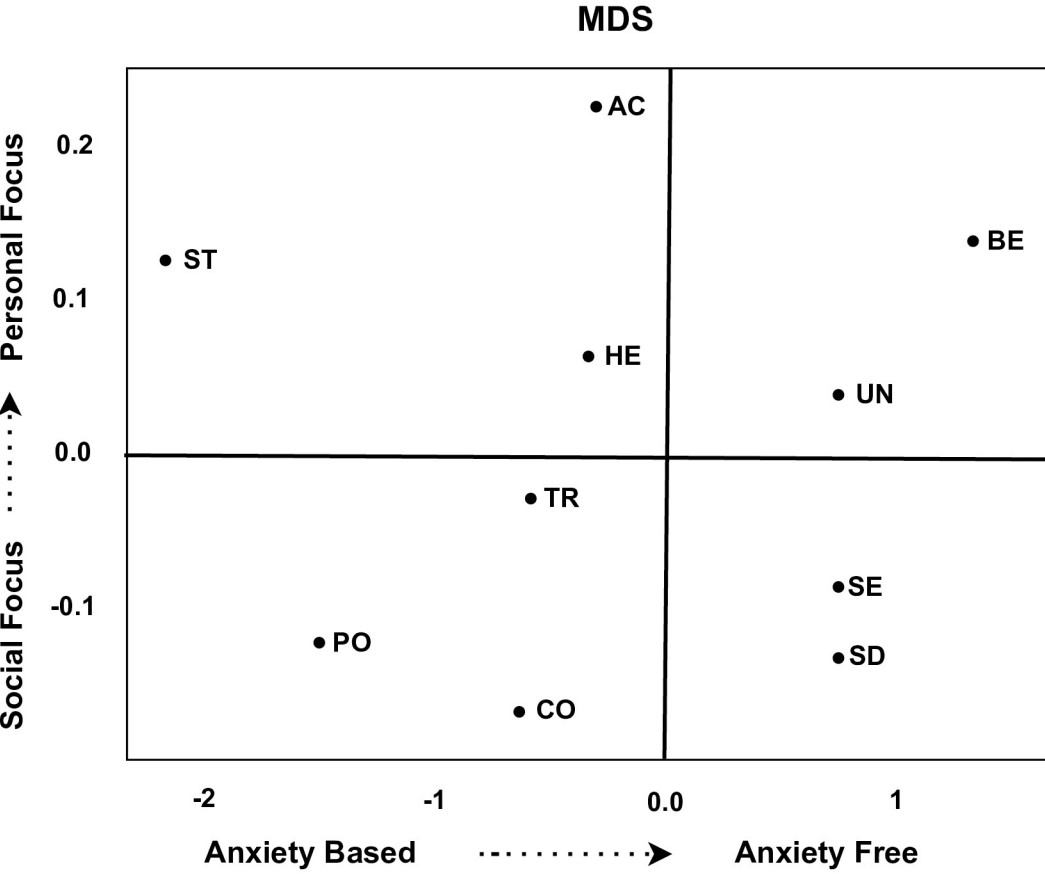

**Fig 4. Dynamic relationship between values for cluster $C_{12}$.**

over time. the Self-direction derives for control and mastery, interactional requirements of autonomy and choosing own goals. The Achievement and Security values also emphasizing competence in terms of prevailing ambitious, successful, and influential life. This is an indication that this cluster focus more on personal interest rather than social characteristics.

## Conclusion

This paper enumerates the changes in basic human values score over time for respondents of SVS survey conducted corss-sectionally every two years since 2001. Each dataset was clustered separately using same clustering algorithm and change detection framework was implemented to these cluster solutions. To remove the effect of older items on the clustering algorithms sliding window model was implemented.

The algorithm detects two clusters that survive through time, out of which one disappeared during time interval (2008, 2012). This cluster have relatively low values score with a gradual increase in personal focused characteristics, such as power, security and achievements. On the other hand, cluster that survive through entire time span became even denser and expand as time progresses. This segment of society have high value score with focusing on social constructs and anxiety free characteristics. Respondents in this cluster assign higher ranks to social qualities such as Universalism, Hedonism, and Benevolence, also prefer to live anxiety free life.

**Table 4. Values scores and MART for cluster $C_{11}$ at respective time points.**

| Window Panes | $D_1$ | $D_2$ | $D_3$ | $D_4$ |
|---|---|---|---|---|
| | (2002) | (2002,2006) | (2004,2008) | (2006,2010) |
| Conformity | .37 | .31 | .26 | .34 |
| | (1.05) | (1.03) | (0.77) | (1.01) |
| Tradition | .08 | .04 | -1.05 | .02 |
| | (1.013) | (0.97) | (0.90) | (0.940) |
| Benevolence | -.78 | -.79 | -1.14 | -.83 |
| | (0.68) | (0.65) | (0.94) | (0.619) |
| Universalism | -.65 | -.62 | -1.09 | -.63 |
| | (0.70) | (0.67) | (0.74) | (0.64) |
| Self-Direction | -.59 | -.54 | .16 | -.53 |
| | (0.75) | (0.75) | (1.05) | (0.75) |
| Stimulation | .83 | .83 | .16 | .85 |
| | (1.02) | (0.99) | (1.05) | (1.00) |
| Hedonism | .11 | .09 | 1.14 | .07 |
| | (0.94) | (0.91) | (0.92) | (0.91) |
| Achievement | .50 | .42 | 1.37 | .39 |
| | (0.91) | (0.92) | (0.94) | (0.92) |
| Power | .94 | .93 | 1.31 | .97 |
| | (0.93) | (0.89) | (0.88) | (0.88) |
| Security | -.48 | .38 | 1.17 | 1.33 |
| | (0.87) | (0.91) | (1.08) | (0.89) |
| Overall | 2.85 | 2.83 | 3.43 | 2.79 |
| | (0.54) | (0.49) | (0.59) | (0.49) |
| $n_i$ | 1447 | 1351 | 926 | 712 |

$(C_{11} \rightarrow C_{22} \rightarrow C_{32} \rightarrow C_{42})$. This cluster survives from time point $t_1$ till time point $t_4$

Likewise, they assign low ranks to self-centered qualities like power, achievements and security.

The values change studies are based on determining the changes in peoples perception about their motivations from past to future. It determines how the actions, thinking, and feelings of the people resist to a shift in the context from the society over time [32, 33]. Results of this study reveals that in last couple of decades universalism is on the rise in European societies. Love for the nature, environment, humanity and kindness to other human beings gain prime importance in their thinking and actions. They give more weight to living an anxiety free life with focusing on socially focused constructs. This was the outcome of a lengthy process in which the concept of human rights acquired a symbolic and politically absolute importance for thinking about how to create a more peaceful world. This has arrived with the construction of welfare states describing essential things such as education, health, security, etc as public services.

## Limitations and future directions

Europe is increasingly characterised by diversity in culture, values, and foreign-born populations. The immigrants belong to different races, ethnic groups and nationalities. Hence, Hofstede's theory could help investigate and quantify basic human values. However, the Schwartz

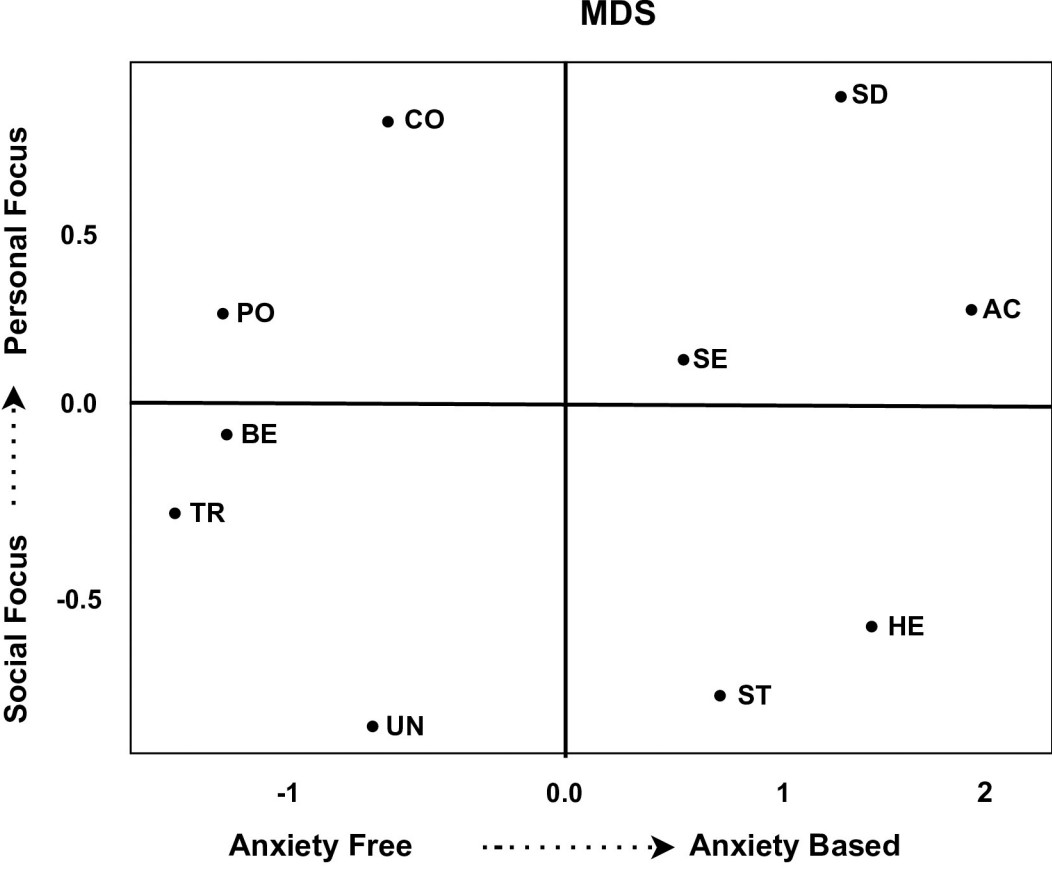

**Fig 5. Dynamic relationship between values for cluster $C_{11}$.**

theory is used because it also explains the relationship between the basic values, which Hofstede's theory could not measure.

Secondly, while this study have demonstrated how values are changing over time in European nations, yet it is difficult to explain precisely why this is happening. There is a need of future research that could address some theoretical and empirical questions that follow from our findings.

## Supporting information

**S1 Fig. The *factoextra()* and *Nbclust()* packages in R-software provides a visualization that can help determining the optimal number of clusters.** These packages provides different options, which also includes the average silhouette statistics. The average silhouette method computes the average silhouette of observations for different values of $k$. The optimal number of clusters $k$ is the one that maximizes the average silhouette over a range of possible values for $k$. For example in the Figure below there are $k = 2$ natural subgroups or clusters in the dataset. (PNG)

**S1 Appendix.**
(PDF)

## Author Contributions

**Conceptualization:** Muhammad Atif.

**Data curation:** Muhammad Farooq.

**Investigation:** Muhammad Waqas.

**Methodology:** Muhammad Shafiq, Gohar Ayub.

**Resources:** Muhammad Waqas.

**Software:** Muhammad Atif.

**Supervision:** Muhammad Atif.

**Writing – review & editing:** Mujeeb Hussain.

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
