## [Decision Letter · Decision Letter 0]

23 May 2022

PONE-D-21-28615Evolution of Basic Human Values Orientations: An Application of Monitoring Changes in Cluster SolutionsPLOS ONE

Dear Dr. Atif,

Thank you for submitting your manuscript to PLOS ONE. After careful consideration, we feel that it has merit but does not fully meet PLOS ONE’s publication criteria as it currently stands. Therefore, we invite you to submit a revised version of the manuscript that addresses the points raised during the review process. In particular, please address both of the reviewers' concerns regarding the discussion of results. Please submit your revised manuscript by Jul 04 2022 11:59PM. If you will need more time than this to complete your revisions, please reply to this message or contact the journal office at plosone@plos.org. Please include the following items when submitting your revised manuscript:A rebuttal letter that responds to each point raised by the academic editor and reviewer(s). You should upload this letter as a separate file labeled 'Response to Reviewers'.A marked-up copy of your manuscript that highlights changes made to the original version. You should upload this as a separate file labeled 'Revised Manuscript with Track Changes'.An unmarked version of your revised paper without tracked changes. You should upload this as a separate file labeled 'Manuscript'.

We look forward to receiving your revised manuscript.

Kind regards,

Hugh Cowley

Senior Editor

PLOS ONE

Journal Requirements:

5. Please upload a new copy of Figures 1, 3 and 4 as the detail is not clear. Please follow the link for more information: https://blogs.plos.org/plos/2019/06/looking-good-tips-for-creating-your-plos-figures-graphics/" https://blogs.plos.org/plos/2019/06/looking-good-tips-for-creating-your-plos-figures-graphics/

Reviewers' comments:

Reviewer's Responses to Questions

**Comments to the Author**

1. Is the manuscript technically sound, and do the data support the conclusions?

Reviewer #1: Yes

Reviewer #2: Partly

2. Has the statistical analysis been performed appropriately and rigorously? 

Reviewer #1: Yes

Reviewer #2: Yes

3. Have the authors made all data underlying the findings in their manuscript fully available?

Reviewer #1: No

Reviewer #2: Yes

4. Is the manuscript presented in an intelligible fashion and written in standard English?

Reviewer #1: Yes

Reviewer #2: Yes

5. Review Comments to the Author

Reviewer #1: In this short article, an analysis is carried out over eight measurement points, over 14 years of basic human values according to Schwartz. The results are very interesting and should be published. Unfortunately, the theoretical background is based on studies that are not very up-to-date. The content of the introduction can be accepted in principle for such a short article, but a more detailed presentation of Schwartz's theory and the meaning of basic human values is missing.

In my opinion, the method has been presented briefly but comprehensibly.

The results should be described in more detail and, above all, better coordinated with the illustrations.

Unfortunately, there is no discussion of the content of the results, which I consider very important.

Introduction

- The short introduction is well structured with the main message that changing basic human values influence human actions and changes therein. But the goal of the study is to "to investigate the dynamics of the relationship between" basic human values. I miss a short theoretical refelection about these value changements.

- At the end of the introduction, the authors write that they want "to achieve the aformentioned objectives". But explicitely, they mention only one goal.

- The literature (of the paper) is not very up-to-date. Apart from technical references (e.g. on R), the references are all before 2014 with one exception (2018). For example, the last paragraph on page 2 begins with "In recent years, one of the most exciting topics is the study of diversity and cultural changes over time." The following sentences are taken from Davidov (2008). That is, by in recent years, the authors are referring to a paper written 14 years ago. Or "Yet recent development" refers to 1992 and 2002.

In fact, there is not much literature on "basic human values" and on "Schwartz Value Survey". But there are recent studies on the topic. The theoretical background should therefore be updated.

Methodology

- Chapter 2 states a slightly different aim of the study: "in order to recognize any variation in value score over time and its impacts on the cultural groups".

- Why do the authors quote Eurostat (2011) when there is new data from 2021 (immigrants in Europe: 5.1% instead of 4.4%)?

- Chapter 2.1: Here I miss the number of participants in the eight surveys or at least a reference to Table 1 and some more information about the samples and the data collection.

- Is there a literature reference for choosing the number of clusters? <- "The choice of an optimal number of clusters was decided by silhouette width, a measure of internal validity. Multidimensional scaling was used for the graphical representation of relation amongst human values in important clusters."

- Very good: R package is avalable.

- The procedure of the analysis has been described in an understandable and comprehensible way. It would be helpful if the authors indicated in the methodology section how the quantity w is obtained.

- The method of multidimensional scaling is not reported.

Results and discussion -> I would rename this section in "Results"

- It would be beneficial for the reader if Figure 1 were described in more detail. Figure 1 contains four illustrations. The text does not explain which figure shows what.

- I suppose the phrase "however Tuckey post hoc test shows that only pairs Also, tuckey post-hoc test show that only significantly different pairs t1, t2, and t3 are different" includes an error. I don't understand it.

- Why do the authors not report the density measures?

- Figures 3 and 4 are not self-speaking and should therefore be better explained in the text. It would also be helpful to write out the scales of the basic human values in the figure instead of using abbreviations.

- "Contrary to cluster C12, Cluster C11 which emerged at t1(2002) and survived till t4(2006, 2010) have [has] relatively lower overall value score ..." ?

- The discussion in this section focuses on the methodological parts, the content of the study is not discussed.

Summary and conclusion -> I would rename this section as "Discussion"; evtl. add a short section "Conclusion"

- The sentence "Mostly the values change studies based on determining the directions in which people’s actions, thinking and feelings have endured transformation." needs some references (or I do not understand it).

- This discussion correctly summarises briefly what was found in the analyses. In the last section I can see a conclusion.

The results (content) are not discussed or linked to theoretical findings and previous studies. I also miss a discussion of the significance of the results. What does it mean for a society to give more space to universalism? How can this be reconciled with the world situation today (especially in Europe)?

- I also miss a discussion on alternative explanations (content-related, methodical). The article does not give any statistical data on the Schwarz Value Survey. The scales each consist of two questions. The fact that there are only two stable clusters could also be due to a lack of reliability or other problems with the scales. Or how stable is the construct of basic human values over time? How stable are human basic values over time?

- Furthermore, the weaknesses and strengths of the study are not addressed.

General

- Language: The English language of the work is easy to read and understand. But there are some stylistic problems. For example, pronouns are regularly omitted where I would use them (see abstract: "[A] change detection algorithm was implemented" or on page 3, chapter 2: "from [the] European Social Survey (ESS), and [the] clusters evolution was traced". In some places in the text this leads to sentences that are difficult to understand; e.g. "To remove the effect of older items on the clustering algorithms [a] sliding window model was implemented."

Reviewer #2: This is an interesting article about values, but I would certainly appreciate a more social and humanistic contribution. Statistical models are important, but only if we can interpret them. Which means that thanks to all statistical operations, something can be explained and understood more deeply.

The article begins with the Introduction section. I understand it should be as short as possible, but the concept of value should be developed. What are the values and how are they understood (definition). Schwartz defines individual values in relation to the goals that motivate actors to act and guide that action. Values are also the source of standards and are ranked in order of importance. Of course, Schwartz's theory is often used, but I'm not sure it's the most popular.

My main concern is Europe - Europe seems to be treated as one 'site', while Europe is very diverse in terms of value, but also in terms of the migration that you mentioned. I would also add the perspective of transformation in the countries of Eastern Europe. Perhaps Hofstede's theory could help investigate this.

While the statistical part is described quite accurately, and I have no comments (besides some limitations in using data from ESS) in the section: result and discussions there is no discussion. What is the answer to your main questions? What are the consequences? What do the results explain?

Even thought in the last part: summing up and concluding some ideas about universalism and livinf anxiety-free life are not enought. I would appreciate elaborating these ideas. What does it mean? And what are the roots of these changes and what are the consequences? What are the differences in Europe. What about migrants?

6. PLOS authors have the option to publish the peer review history of their article (what does this mean?). If published, this will include your full peer review and any attached files.

Reviewer #1: No

Reviewer #2: No

---

## [Author Response · Author response to Decision Letter 0]

6 Jul 2022

Dear Editor and Reviewers, 

We would like to thank Reviewers for taking the necessary time and effort to review the manuscript. We sincerely appreciate all your valuable comments and suggestions, which helped us in improving the quality of the manuscript.

1. The Manuscript is updated to the template of PLOS ONE style.

2. The Figures are updated. 

3. The data is openly available from the ESS repository. 

Response to the Reviewers is uploaded in a separate file.

---

## [Decision Letter · Decision Letter 1]

8 Aug 2022

PONE-D-21-28615R1Evolution of Basic Human Values Orientations: An Application of Monitoring Changes in Cluster SolutionsPLOS ONE

Dear Dr. Atif,

Thank you for submitting your manuscript to PLOS ONE. After careful consideration, we feel that it has merit but does not fully meet PLOS ONE’s publication criteria as it currently stands. Therefore, we invite you to submit a revised version of the manuscript that addresses the points raised during the review process. Two concerns noted in the previous round of review were not addressed satisfactorily: A) The following comment was not included in your response-to-reviewers: "My main concern is Europe - Europe seems to be treated as one 'site', while Europe is very diverse in terms of value, but also in terms of the migration that you mentioned. I would also add the perspective of transformation in the countries of Eastern Europe. Perhaps Hofstede's theory could help investigate this." Please respond to this concern by discussing this in your manuscript and by noting this as a limitation in your Results/Discussion section. B) The reference to Eurostat was updated in the tracked changes version of the manuscript, but not the clean copy. Please ensure the content of the two documents is identical.

We look forward to receiving your revised manuscript.

Kind regards,

George Vousden

Staff Editor

PLOS ONE

Journal Requirements:

Reviewers' comments:

Reviewer's Responses to Questions

**Comments to the Author**

1. If the authors have adequately addressed your comments raised in a previous round of review and you feel that this manuscript is now acceptable for publication, you may indicate that here to bypass the “Comments to the Author” section, enter your conflict of interest statement in the “Confidential to Editor” section, and submit your "Accept" recommendation.

Reviewer #1: All comments have been addressed

2. Is the manuscript technically sound, and do the data support the conclusions?

Reviewer #1: Yes

3. Has the statistical analysis been performed appropriately and rigorously? 

Reviewer #1: Yes

4. Have the authors made all data underlying the findings in their manuscript fully available?

Reviewer #1: Yes

5. Is the manuscript presented in an intelligible fashion and written in standard English?

Reviewer #1: Yes

6. Review Comments to the Author

Reviewer #1: The paper has improved greatly after the revision. Most of the reviewers' suggestions have been implemented.

The analysis of values in Europe over time and the corresponding results are valuable and informative and should be published.

Unfortunately, an important shortcoming remains: While the results and tables have been better described in the results/discussion section, there is no discussion of the results in this paper in comparison to existing literature on values and values in Europe. The results are also not placed in the political environment of Europe. For example, there has been a strengthening of environmental policy in recent years, which fits well with the cluster of less individualism. At the same time, the right has made strong gains across Europe. This may also correspond to the same cluster: more responsibility to the state, less to the individual. But do both fit the two clusters? Where is the diffusion of responsibility reflected? And so on.

7. PLOS authors have the option to publish the peer review history of their article (what does this mean?). If published, this will include your full peer review and any attached files.

Reviewer #1: No

---

## [Author Response · Author response to Decision Letter 1]

20 Aug 2022

Thank you for giving me the opportunity to submit a revised draft of my manuscript titled Evolution of Basic Human Values Orientations: An Application of Monitoring Changes in Cluster Solutions to PLOS ONE. We appreciate the time and effort that you and the reviewers have dedicated to providing your valuable feedback on our manuscript. We are grateful to the reviewers for their insightful comments on my paper. 

Here is a point-by-point response to the reviewers’ comments and concerns.

A) The following comment was not included in your response-to-reviewers: "My main concern is Europe - Europe seems to be treated as one 'site', while Europe is very diverse in terms of value, but also in terms of the migration that you mentioned. I would also add the perspective of transformation in the countries of Eastern Europe. Perhaps Hofstede's theory could help investigate this."

Response: This is added to the manuscript as limitations of the study. 

B) The reference to Eurostat was updated in the tracked changes version of the manuscript, but not the clean copy. Please ensure the content of the two documents is identical.

Response: The reference in the clean copy and track changes copy is updated, but it was not highlighted in the track changes copy. It is highlighted in track changes copy.

---

## [Editor Report · Decision Letter 2]

1 Sep 2022

Evolution of Basic Human Values Orientations: An Application of Monitoring Changes in Cluster Solutions

PONE-D-21-28615R2

Dear Dr. Atif,

We’re pleased to inform you that your manuscript has been judged scientifically suitable for publication and will be formally accepted for publication once it meets all outstanding technical requirements.

Kind regards,

George Vousden

Deputy Editor in Chief

PLOS ONE

Additional Editor Comments (optional):

Please provide references to the newly added paragraph underneath the subtitle 'Limitations and future directions' (line 291).
---

## [Editor Report · Acceptance letter]

12 Sep 2022

PONE-D-21-28615R2 

Evolution of basic human values orientations: An application of monitoring changes in cluster solutions 

Dear Dr. Atif:

I'm pleased to inform you that your manuscript has been deemed suitable for publication in PLOS ONE. Congratulations! Your manuscript is now with our production department. 

Kind regards, 

on behalf of

Dr. George Vousden 

Staff Editor

PLOS ONE